# Study of the Soft Magnetic Properties of FeSiAl Magnetic Powder Cores by Compounding with Different Content of Epoxy Resin

**DOI:** 10.3390/ma16031270

**Published:** 2023-02-02

**Authors:** Zhengqu Zhu, Jiaqi Liu, Huan Zhao, Jing Pang, Pu Wang, Jiaquan Zhang

**Affiliations:** 1School of Metallurgical and Ecological Engineering, University of Science and Technology Beijing, Beijing 100083, China; 2Qingdao Yunlu Advanced Materials Technology Co., Ltd., Qingdao 266232, China

**Keywords:** FeSiAl magnetic powder cores, epoxy resin, loss separate, soft magnetic properties

## Abstract

FeSiAl is a commonly used soft magnetic material because of its high resistivity, low core loss, and low cost. In order to systematically study the effect of epoxy resin (EP) on the insulated coating and pressing effect of FeSiAl magnetic powders, six groups of composite powders and their corresponding soft magnetic powder cores (SMPCs) were prepared by changing the content of EP, and the soft magnetic properties of the powders and SMPCs were characterized. The results showed that FeSiAl powders exhibited good sphericity and morphology. The *M*_s_ of FeSiAl/EP composite powders was between 117.4–124.8 emu·g^−1^ after adding (0.3, 0.5, 0.7, 1, 1.5, and 2 wt. %) EP. The permeability *μ_e_* of SMPCs increased first and then decreased with the increase in EP content. Among them, when the EP content was 1 wt. %, the corresponding SMPCs had the highest *μ_e_* and excellent DC bias performance (63%, 100 Oe). In the whole test frequency range (50~1000 kHz), SMPCs with 1 wt. % EP content had the lowest core loss (1733.9 mW·cm^−3^ at 20 mT and 1000 kHz). After that, the loss separation study in the low-frequency range (50~250 kHz) was conducted, and the hysteresis loss and eddy current loss of SMPCs with 1 wt. % EP content were also the lowest. In addition, SMPCs also exhibited the best overall performance when the EP content was 1 wt. %. The results of this study can guide the design of composite insulation coating schemes and promote the development of soft magnetic materials for medium and high frequency applications.

## 1. Introduction

Soft magnetic powder cores (SMPCs) are a kind of soft magnetic composite made by pressing magnetic powders coated with insulating materials. They exhibit great potential in downstream fields such as photovoltaics, energy storage, variable frequency air conditioning, new energy vehicles, charging piles, uninterrupted power supplies (UPS), 5G base stations, and servers due to their excellent soft magnetic properties, flexible shape, and easy processing [1,2,3].

Insulated coating and pressing are the keys to preparing SMPCs. How to minimize the eddy current loss between magnetic powder cores and ensure the pressing effect of composite magnetic powder cores has always been a research hotspot. In recent years, some scholars have deposited inorganic insulation layers such as SiO_2_, TiO_2_, and Al_2_O_3_ [4,5,6,7,8,9] on the surfaces of magnetic powders by the sol–gel method and prepared ferrite insulation layers such as MnZn and NiZn [10,11,12,13] by the coprecipitation method to obtain SMPCs with excellent comprehensive performance. However, these two methods have many drawbacks, such as a complex coating process, high production cost, and unstable insulation properties. Additionally, most of the above-mentioned coatings are limited to small-scale research in the laboratory, and it is difficult to promote them to industrial mass production.

Epoxy resin (EP) is an organic insulating agent with both insulation and binding properties. It is widely used in the preparation process of SMPCs because it is easy to adhere to the surfaces of particles to achieve a uniform coating [14]. Based on the sol–gel method, Zhou et al. [15] prepared TiO_2_ insulating layers on the surfaces of the magnetic powders by a hydrolysis condensation reaction and added 2 wt. % EP as a binder to improve the compressibility of FeSiBCCr/TiO_2_ SMPCs. Chi et al. [16] used FeSiBCP amorphous powders as raw materials and prepared Fe_3_O_4_ nano-scale insulation layers by hydrothermal reaction in an alkaline environment and then embedded EP into the Fe_3_O_4_ coating matrix by mechanical mixing, which further improved the structural uniformity of composite coatings and the high-frequency comprehensive performance of SMPCs. Liu et al. [17] prepared FeSiBCCr@phosphate@EP core–shell structured SMPCs with excellent soft magnetic properties by using EP as the second insulation layer after the magnetic powders were passivated by phosphoric acid, which further reduced the core loss (1118.7 mW·cm^−3^ at 20 mT and 1000 kHz). It can be seen that the organic insulating agent EP can effectively enhance the compressibility of SMPCs and reduce the eddy current loss; thus, it is commonly used as the main or secondary insulating material.

The FeSiAl powders produced by gas atomization have high sphericity, high permeability, low coercivity, high resistivity, and good wear resistance. Additionally, the magnetostriction coefficient and magnetic anisotropy constant of the FeSiAl alloy both tend to almost zero, so it is a kind of magnetic powder with excellent soft magnetic properties [18]. However, there are few systematic studies on the preparation of FeSiAl SMPCs only using EP as the insulating agent, and the relationship between the optimum addition amount and the characteristics of FeSiAl powder is still unknown. Therefore, in this paper, FeSiAl@EP SMPCs were prepared by using EP as the insulating agent, and the effects of EP content on the coating effect and soft magnetic properties of SMPCs were studied by various means to investigate the influence mechanism, which could provide the theoretical basis and practical guidance for the development and design of a composite insulated coating scheme.

## 2. Materials and Methods

### 2.1. Preparation of FeSiAl Powders

The experimental raw materials used in this study were FeSiAl crystalline powders (Qingdao Yunlu Advanced Materials Technology Co., Ltd, Qingdao, China) prepared by the gas atomization process. The composition was 8.7 wt. % Si, 5.6 wt. % Al, and 85.7 wt. % Fe. Epoxy resin was purchased from Shandong Huijia Magnetoelectricity Technology Co., Ltd. in Dezhou, China.

### 2.2. Preparation of SMPCs

Firstly, EP solution was prepared with 0.3, 0.5, 0.7, 1, 1.5, and 2 wt. % solute EP and 10 wt. % solvent acetone, respectively. The 2.5 kg FeSiAl powders were poured into the mixer, and the solution was poured after stirring for 30 s at room temperature. The heating temperature was 100 °C, and the stirring was 1~2 h until the powders were uniformly coated. The coated powders were sieved through a standard sieve of –140 mesh and were named P1~P6 according to the EP content. The powders (with the mass of 2 g) were compacted into annular SMPC samples of Φ 14 mm × Φ 8 mm × h 3.1 mm by two-way pressing with a pressure of 1800 MPa at room temperature. Subsequently, the annular SMPCs were annealed in vacuum at 200 °C for 1 h to release the internal stress caused by pressing and to enhance the strength of SMPCs. The specific preparation process is shown in Figure 1. SMPCs corresponding to P1~P6 composite powders were named S1–S6.

### 2.3. Characterization

The microscopic morphology of FeSiAl raw powders and coated composite powders was observed by scanning electron microscopy (SEM, Phenom Pro Desktop SEM, Phenom-World BV, Eindhoven, The Netherlands). The particle size of FeSiAl powders was measured by a laser particle size analyzer (BT-9300S, Bettersize Instruments Ltd., Dandong, China). The phase of each sample was analyzed by X-ray diffraction (XRD, D2 PHASER, BRUKER AXS, Karlsruhe, Germany) using Cu Kα radiation with λ = 0.15406 nm. The step size was 0.02°, the scanning range was 30°~90°, the tube voltage was 30 kV, and the tube current was 10 mA. The saturation magnetization of each sample was measured by a vibrating sample magnetometer (VSM, Lake Shore 8604, Lake Shore Cryotronics, Inc., Westerville, OH, USA). The measurement range of saturation magnetization was ± 10,000 Oe, and the step size was 200 Oe. The DC bias performance of each sample at 100 kHz and 1 V and the effective permeability *μ_e_* within the frequency range of 10 kHz~1 MHz were tested using the TH2816B/TH2826 LCR tester. The density of the SMPCs was measured according to the Archimedes drainage method. The coercivity of SMPCs was measured by a TD8220 soft magnetic DC tester. The resistivity of SMPCs was obtained by a four-probe method using an ST2742B automatic powder resistivity tester. The core loss of SMPCs was measured by a B–H analyzer (IWATSU-SY-8219) in the frequency range of 50 to 250 kHz. The maximum magnetic flux densities (*B_m_*) were 0.01 T, 0.03 T, and 0.05 T, respectively. In addition, the core loss *P_cv_* of SMPCs was also measured at *B_m_* = 0.02 T and *f* = 100 kHz~1 MHz.

The effective permeability was calculated by the following equation [17]:(1)$$\mu_{e}=\frac{Ll_{e}}{\mu_{0}N^{2}A_{e}}$$
where *L* is the inductance, *l_e_* is the mean flux density path length, *N* is the number of coil turns, *A_e_* is the cross-section area of the SMPCs, and *μ*_0_ is the permeability of vacuum, 4π × 10^−7^ H/m.

## 3. Results

### 3.1. Characterization of FeSiAl Powders

Figure 2 is the XRD pattern of FeSiAl raw powder P0 and other composite powders P1~P6. It can be seen from Figure 2 that all the XRD patterns of the powders show three distinct sharp diffraction peaks. According to the Scherrer formula, the average grain size of FeSiAl powder is 22.4 nm. Among them, the peak intensity of P0 is the highest, and the peak intensity of P1~P6 decreases in turn, indicating that the content of the crystal phase in the composite powders gradually decreases with the increase in EP content. The three diffraction peaks are located at 44.73°, 65.21°, and 82.69°, respectively, and Miller indices are (220), (400), and (422), respectively, which are the characteristic peaks of the typical FCC structure Al_0.3_Fe_3_Si_0.7_ crystal phase [19]. EP is a kind of organic polymer compound, so EP is an amorphous material that shows a broad diffuse halo in XRD. However, the EP content added in this study is less, and the corresponding peak is overlapped by the diffraction peak of the Al_0.3_Fe_3_Si_0.7_ crystal phase; thus, the EP is not detected by XRD in the five samples.

Figure 3 is the particle size distribution of FeSiAl powders. It can be seen from Figure 3 that the powder particle size deviates from the normal distribution with a *d*_50_ of 45.86 μm, exhibiting a generally coarse particle size. As can be seen from the SEM image in the inset of Figure 3, the FeSiAl powders show high sphericity and smooth surfaces.

The SEM images shown in Figure 4 P0–P6 show the evolution of the surface morphology of FeSiAl powders under different EP contents. It can be seen that with the increase of EP content, the surface of the powders is gradually dim, and small particles adhere to the surfaces of large particles. Among them, obvious EP agglomerations on the surface of P5 and P6 can be observed, while there is less for P4. This indicates that when the EP content exceeds 1 wt. %, the powder surface is unevenly coated, resulting in agglomerations of EP. It is worth mentioning that by controlling the amount of EP added, small particles adhere to the surface of large particles, which is beneficial to reduce the air gaps between particles in the subsequent pressing process and improve the compressibility and density of SMPCs.

Figure 5 shows the hysteresis loops of FeSiAl powders and composite powders P1~P6 measured at room temperature. It can be seen from the enlarged view that the *M*_s_ of each sample is not consistent. Among them, the *M*_s_ of the raw powders is the highest (127.3 emu·g^−1^), while the *M*_s_ of the coated powders is between 117.4~124.8 emu·g^−1^, which is lower than that of the raw powders. This decrease indicates that the presence of the non-magnetic insulating layers occupies a certain volume fraction, resulting in a decrease in the *M*_s_ of the composite powders under the magnetic dilution effect [20].

### 3.2. Performance Characterization of SMPCs

The frequency dependence of the effective permeability *μ_e_* of S1 to S6 is shown in Figure 6a. In the frequency range of 10 kHz to 1 MHz, the *μ_e_* of the 6 SMPCs decreases with increasing frequency. Among them, S1 shows the worst frequency stability because of the thinnest insulating layers. When the EP content increases from 0.5 to 2.0 wt. %, the thickness of the insulating layers increases, and the stability of the permeability of the corresponding SMPCs at high frequency is also improved.

It can be seen from Figure 6b that the *μ_e_* of SMPCs increases from 33.9 to 38.2 with the increase of EP content from 0.3 to 1.5 wt. %, but when the EP content continues to increase to 2.0 wt. %, the permeability decreases to 36.7. The permeability of SMPCs increases first and then decreases with the increase of EP content. It is well known that the variation of *μ_e_* is closely related to the density of SMPCs, the air gaps, and the microstructure of the insulating layers. Therefore, in this paper, the Archimedes principle is applied to calculate the density of SMPCs. Firstly, the mass w_1_ of the sample in air and the mass w_2_ of the sample after being hung on the wire are measured. Then the sample is put into water and the mass w_3_ is measured. The volume of the sample is calculated by the equal volume method. Finally, the density is calculated according to Equation (2) as follows:(2)$$\rho =\frac{w_{1}\rho_{1}}{\left(w_{2}-w_{3}\right)}$$
where *ρ* is the sample density, g∙cm^−3^, and *ρ*_1_ is the density of water used for weighing, which is 1.00 g∙cm^−3^. As shown in Figure 6b, it can be seen that the variation trend of *μ_e_* is consistent with density. It can be speculated that when the EP content increases from 0.3 to 1.5 wt. %, small particles adhere to the surfaces of large particles, and the air gaps between particles decrease, which improves the compressibility of SMPCs and the volume fraction of the magnetic materials, resulting in an increase in *μ_e_*. When the non-magnetic insulating material is excessive, the powder coating is uneven, the air gaps increase, and the demagnetization field in the magnetic powder cores increases, resulting in a decrease in *μ_e_* [21].

The DC bias is defined as the ratio of the permeability of SMPCs in the applied magnetic field to that without the magnetic field. The DC bias of the samples is shown in Figure 7. The DC bias performance of S1~S6 shows an overall attenuation trend with the increase of the applied magnetic field. Especially when the magnetic field intensity is greater than 10 Oe, the percent permeability begins to drop sharply, from 99% at 10 Oe to about 64% at 100 Oe. Theoretically, the permeability of SMPCs is directly related to the volume fraction of air gaps. Although the air gaps in SMPCs play a role in pinning the magnetic domain walls during the magnetization process and hinder the attenuation of the permeability, the density of SMPCs also decreases due to the air gaps, resulting in a decrease in the permeability [22]. Therefore, under the same magnetic field intensity, the lower the permeability of SMPCs, the stronger the ability to resist saturation attenuation, that is, the better the DC bias performance, as can be seen from the inset in Figure 7.

Figure 8a is the core loss versus frequency curve of SMPC samples. It can be found that with the increase in the EP content, the core losses decrease first and then increase. Among them, the core losses of S4 are the lowest, and the core losses of other samples are relatively close, as shown in Figure 8b. The role of the EP is to increase the resistivity of the SMPCs by uniformly coating the particles with insulation layers, thereby greatly reducing core losses, but the content of EP should not be excessive.

The core loss of magnetic devices occupies a large proportion of switching between power supply and motor, so it is necessary to study the core losses of SMPCs. Figure 9a shows the relationship between *P_cv_* and magnetic induction intensity for 6 samples at *f* = 100 kHz, while Figure 9b shows the relationship between *P_cv_* and frequency for 6 samples at *B_m_* = 50 mT. The core losses of S1~S6 increase with the increase in magnetic induction intensity and frequency. It can be seen from Figure 9b that, in the range of 50~100 kHz, the *P_cv_* of samples is relatively close, but as the frequency continues to increase, the *P_cv_* of the 6 samples begins to show a significant difference, indicating that the SMPCs prepared in this study are suitable for medium and high frequencies. In addition, when the EP content is increased from 0.3 to 1 wt. %, the *P_cv_* showed a decreasing trend, whereas when the EP content further increased from 1 to 2 wt. %, *P_cv_* increased slightly. Among them, S4 shows the lowest core loss of 512 mW·cm^–3^ (at 100 kHz and 50 mT).

In order to further explore the influence mechanism of EP content on the core loss of SMPCs, the loss separation of *P_cv_* for 6 samples was studied. In theory, the core loss *P_cv_* can be expressed as [23]:(3)$$P_{cv}=P_{h}+P_{e}+P_{\mathrm{ex}}$$
where *P*_h_, *P*_e_, and *P*_ex_ are hysteresis loss, eddy current loss, and residual loss, respectively. Among them, the residual loss *P*_ex_ is mainly caused by the magnetization relaxation process or dispersion. It is a frequency-independent constant under a low-frequency weak magnetic field, and its value is generally small. Compared with *P*_h_ and *P*_e_, *P*_ex_ is negligible. Therefore, *P*_ex_ is ignored in the loss separation analysis of this paper. Figure 9c,d show the curves of *P*_h_ and *P*_e_ with frequencies. With the increase in EP content, *P*_h_ and *P*_e_ decrease first and then increase. *P*_h_ refers to SMPCs in an alternating magnetic field due to the repeated magnetization hysteresis phenomenon caused by the consumption of energy, which can be expressed as follows:(4)$$P_{h}=C_{h}\cdot B_{m}{}^{\alpha }\cdot f$$
where *C*_h_ is the hysteresis coefficient, *B_m_* is the magnetic induction intensity, *f* is the frequency, and *α* is the simulation coefficient. *C*_h_ is related to *H*_c_; the lower value of *C*_h_ represents a lower *P*_h_. Figure 9e shows the *C*_h_ and *H*_c_ of FeSiAl@EP SMPCs with different EP content. With the gradual increase in EP content, *H*_c_ decreased from 0.801 Oe to 0.664 Oe, and then increased to 1.046 Oe, while *C*_h_ decreased from 418 to 334, and then increased to 402.8, which is consistent with the changing trend of *P*_h_ in Figure 9c.

The eddy current loss *P*_e_ refers to the energy loss caused by the induced current generated by SMPCs in the alternating magnetic field. *P*_e_ can be expressed as [24]:(5)$$P_{e}=C_{e}\cdot B_{m}{}^{2}\cdot f^{2}=C\cdot B_{m}{}^{2}\cdot f^{2}\cdot d^{2}/\rho$$
where *C_e_* is the eddy current coefficient, *C* is the proportional constant, *d* is the diameter of composite magnetic powders, and *ρ* is resistivity. It can be seen that *P*_e_ is proportional to the square of the diameter of the magnetic powders and inversely proportional to the resistivity of SMPCs; that is, the size of the particles should also be taken into account while coating the powders to reduce the eddy current path. It can be seen from Figure 9f that the resistivity of the composite magnetic powders increases continuously with the increase in EP content, which is not consistent with the changing trend of *P*_e_ for FeSiAl@EP SMPCs. This is because the agglomeration of the particles occurs due to the excessive amount of EP (1.5~2 wt. %), reducing the local resistivity of SMPCs, as shown in Figure 4f,g. The uneven coating causes the eddy current path of the particles inside the SMPCs to become relatively large, resulting in an abnormal increase in *P*_e_. In addition, the agglomeration of the particles further increases the diameter *d* of the magnetic powders, resulting in the negative effect of increasing eddy current loss beyond the positive effect of decreasing eddy current loss caused by the increased resistivity, so *P*_e_ will increase inversely with the increase in EP. Therefore, the content of the insulating agent EP should be controlled within a proper range. It can be seen from the hysteresis loss in Figure 9c and the eddy current loss in Figure 9d that the core losses are the lowest when the EP content is 1 wt. %, which also shows that the insulation coating effect of the S4 sample in this study is the best.

Based on the discussion of all the above experimental results, considering the properties of saturation magnetization, permeability, DC bias performance, and core losses of the 6 samples, it is considered that the SMPC with the best comprehensive performance in the high-frequency range is sample S4 with 1 wt. % EP. Table 1 summarizes the soft magnetic properties of FeSiAl@EP SMPCs prepared in this study. It can be seen from the table that the samples in this study show high saturation magnetization (117.4~124.8 emu·g^−1^) and low core losses under different conditions (512.0~630.9 mW·cm^–3^ at 50 mT and 50 kHz; 1733.9~2110.3 mW·cm^–3^ at 20 mT and 1 MHz). The samples also have a high permeability (33.9 to 38.2) and show excellent stability in the high-frequency range. In addition, the excellent DC bias performance (62.7 to 66.5, 100 Oe) of SMPCs meets the requirements of high-end inductors and transformer coils. The improvement of saturation magnetization, permeability, and core losses also expands the application range of SMPCs into the low-frequency, high-frequency, and high current fields. It is worth mentioning that the insulating agent used in this paper is only EP, and its insulation effect is not the best, resulting in relatively high core losses. However, after the improvement of the subsequent coating process, it is expected to further improve the comprehensive performance of SMPCs, especially the core losses.

## 4. Conclusions

In order to explore the influence mechanism of organic insulating agent EP on the soft magnetic properties of FeSiAl SMPCs, the properties of FeSiAl composite magnetic powders with different EP contents and the corresponding FeSiAl@EP SMPCs were tested and characterized by various means. The main conclusions are as follows:With the increase in EP content, the permeability of SMPCs increased from 33.9 to 38.2, and then decreased to 36.7, which showed a trend of increasing first and then decreasing. Six SMPCs had high DC bias performance, between 62~67% (100 Oe).The loss separation analysis of SMPCs in the range of 50~250 kHz showed that when the EP content was 1 wt. %, the coating of sample S4 was the most uniform, resulting in the lowest coercivity and hysteresis losses. The smaller magnetic powder particles also curbed the deterioration of eddy current losses and thus obtained the lowest core losses.In the high-frequency range, due to uniform EP coating and high density of SMPCs, S4 exhibited the lowest core losses (1733.9 mW·cm^−3^ at 20 mT and 1000 kHz), high permeability (36.1), and excellent DC bias performance (63%, 100 Oe).

The FeSiAl@EP SMPCs with EP content of 1 wt. % prepared in this study have excellent comprehensive soft magnetic properties, and the core losses can be further reduced by the improvement of the subsequent coating process, which is helpful to the development and design of FeSiAl insulated coating schemes and the preparation of high-performance FeSiAl SMPCs.

## Figures and Tables

**Figure 1 materials-16-01270-f001:**
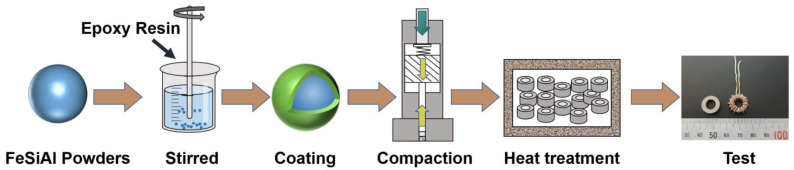
Schematic diagram of fabrication process for FeSiAl SMPCs.

**Figure 2 materials-16-01270-f002:**
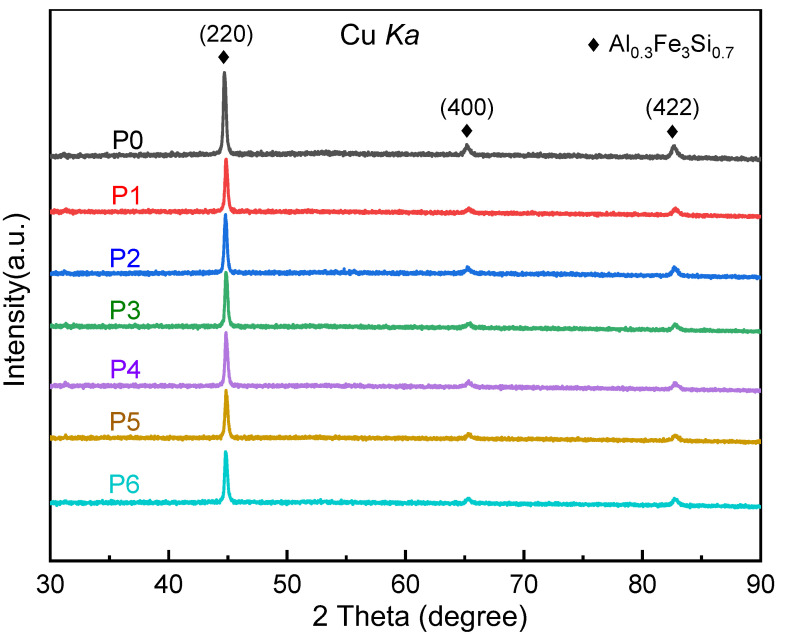
XRD patterns of FeSiAl powders P0~P6.

**Figure 3 materials-16-01270-f003:**
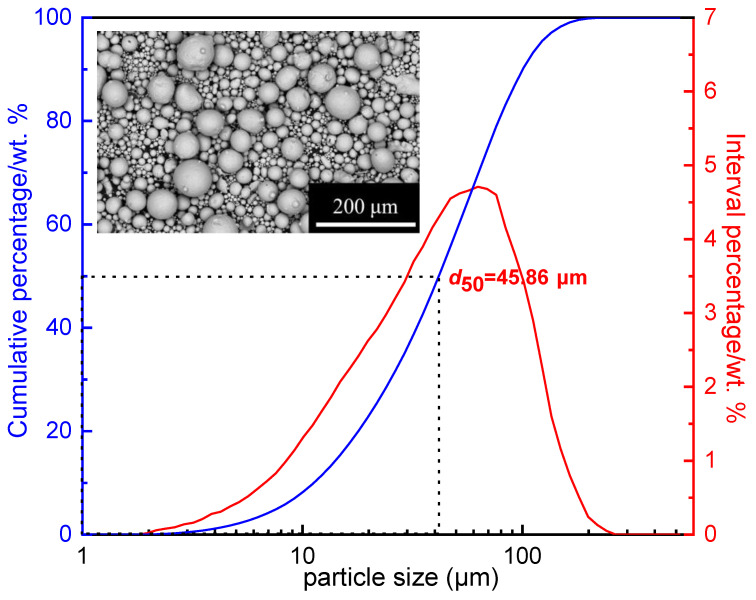
Size distribution of FeSiAl raw powders; the inset is the SEM image of FeSiAl powders.

**Figure 4 materials-16-01270-f004:**
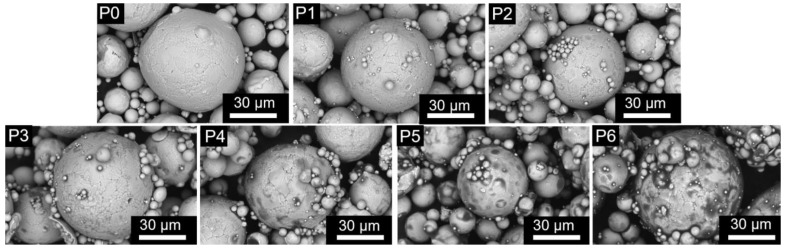
SEM images of FeSiAl powders P0~P6.

**Figure 5 materials-16-01270-f005:**
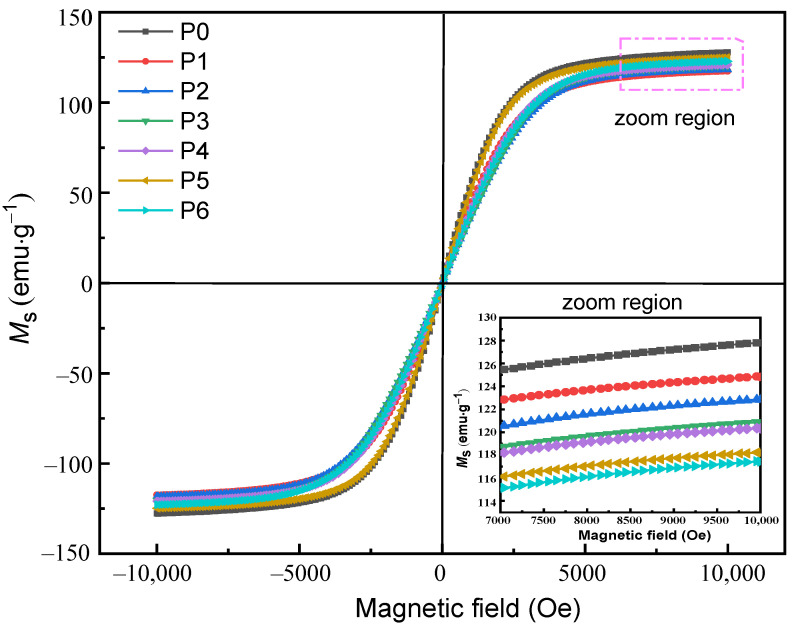
The hysteresis loops of powders P0~P6.

**Figure 6 materials-16-01270-f006:**
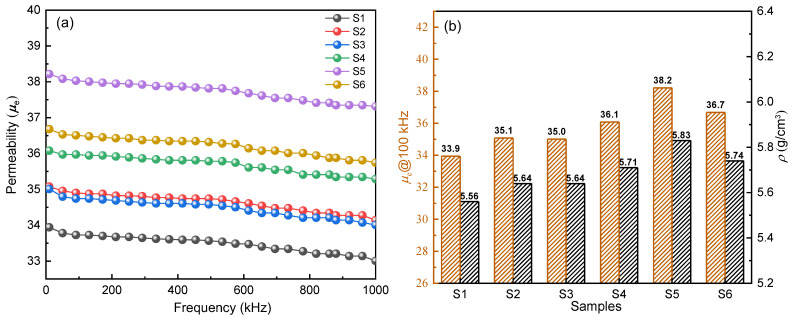
(**a**) Variations in the effective permeability with frequency for S1~S6; (**b**) *μ_e_*@100 kHz and density for S1~S6.

**Figure 7 materials-16-01270-f007:**
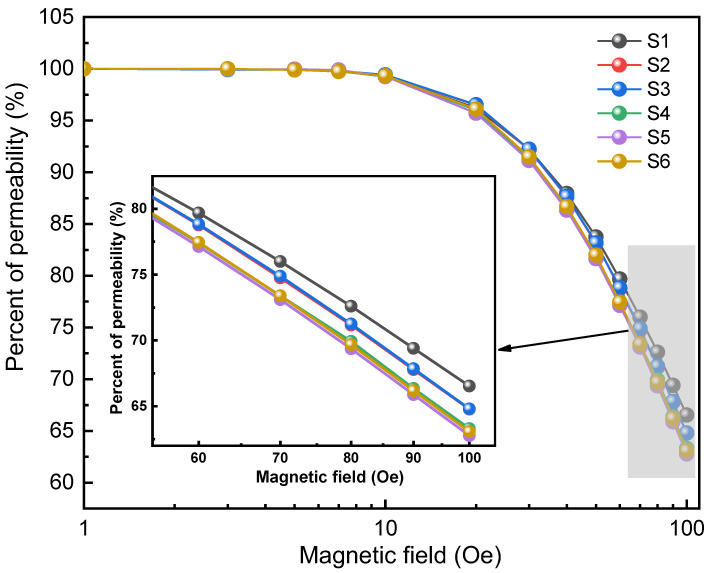
DC bias field dependence for SMPCs S1~S6 at magnetic field of 1~100 Oe.

**Figure 8 materials-16-01270-f008:**
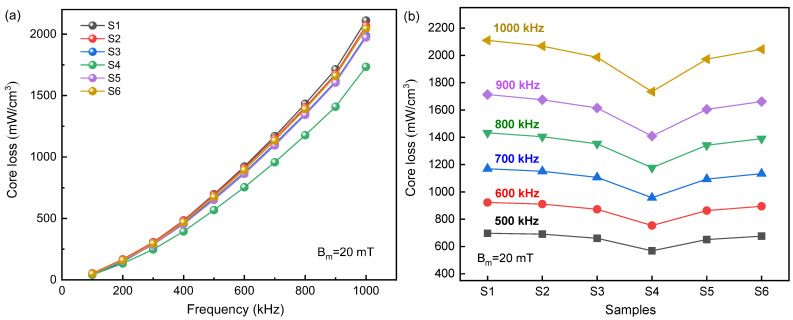
The total core losses of SMPCs for S1~S6:(**a**) under different EP content; (**b**) in the range of 500~1000 kHz.

**Figure 9 materials-16-01270-f009:**
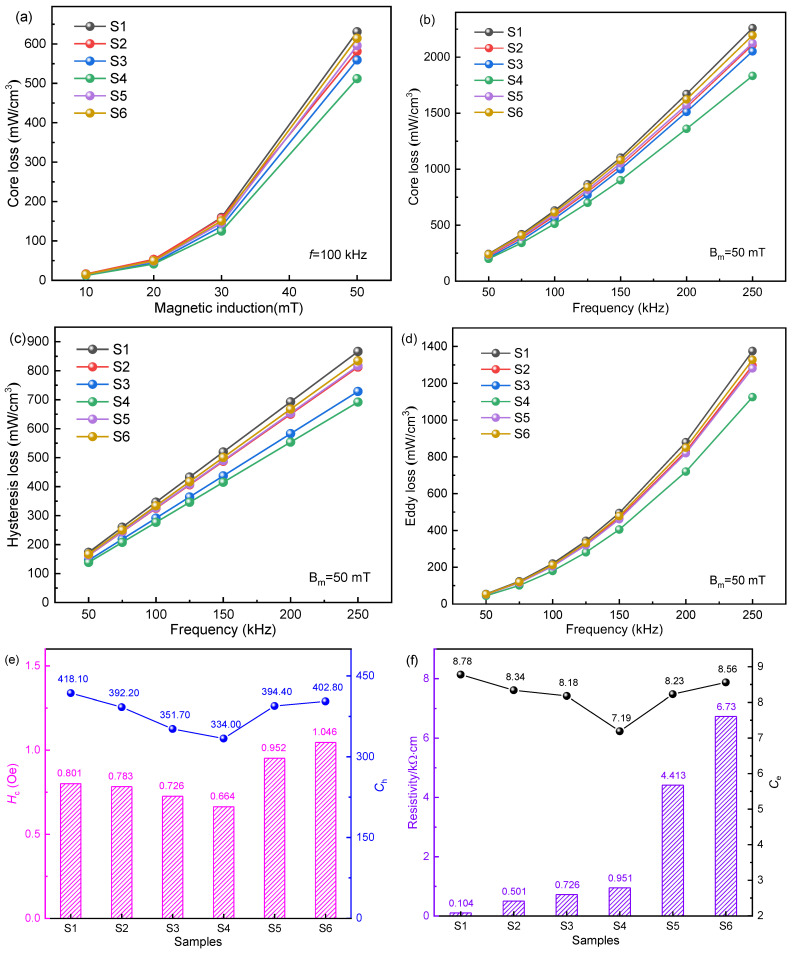
(**a**) Simulation results for frequency dependence of the core losses of S1~S6 under different magnetic induction. *P_cv_* (**b**), *P*_h_ (**c**), and *P*_e_ (**d**) of FeSiAl@EP SMPCs with different frequencies. (**e**) The coercivity and the coefficient of hysteresis loss of the samples. (**f**) The resistivity and the coefficient of eddy current loss of the samples of S1~S6.

**Table 1 materials-16-01270-t001:** Comparison of soft magnetic properties of SMPCs prepared in this work.

Sample	*M* _s_	*ρ*	*μ_e_*	Core Loss, *P_cv_* (mW·cm^–3^)			DC-Bias (%)
	emu∙g^−1^	g∙cm^−3^	100 kHz	50 kHz/50 mT	100 kHz/50 mT	1 MHz/20 mT	100 Oe
S1	117.4	5.56	33.9	243.7	630.9	2110.3	66.5
S2	118.2	5.64	35.1	224.9	581.1	2068.1	64.7
S3	120.3	5.64	35.0	210.0	559.3	1985.8	64.8
S4	120.8	5.71	36.1	199.3	512.0	1733.9	63.3
S5	124.8	5.83	38.2	233.9	595.7	1972.5	62.7
S6	122.8	5.74	36.7	239.7	614.4	2044.5	63.1

## Data Availability

The data presented in this study are available on request from the corresponding author.

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
