# Peer review of "Study of the Soft Magnetic Properties of FeSiAl Magnetic Powder Cores by Compounding with Different Content of Epoxy Resin"

_materials, 2023, doi:10.3390/ma16031270_

Round 1

Reviewer 1 Report

Submitted manuscript entitled “Study of the soft magnetic properties of FeSiAl magnetic powder cores by compounding with different content of epoxy resin” deals with the subject of the investigation of the magnetic properties of sift magnetic materials.

The Authors claim that such materials used for fabrication of soft magnetic powder coils are interesting as they can be used in “photovoltaics, energy storage, variable frequency air conditioning, new energy vehicles, charging piles, uninterrupted power supply (UPS), 5G base stations and servers”. However there is no reference given to support this statement. References 1-3 mentioned in this same sentence refer to properties and/or fabrication studies rather than real applications descriptions. If there is any reference to prove that the real-life application exist please provide it (patent, published studies, etc.). Otherwise I suggest to use a term like “potential applications”.

There are some sentences in the text that are not grammatically correct (e.g. they lack a predicate). Please give special attention to “conclusions” and “characterization” chapters in that sense.

Figure 3: please extend the caption – explain the exes, curves, inset.

Figure 4: Putting sample names over the SEM images would make it much easier for the reader. Could the authors show the microstructure of the pressed samples. I understand that those are the important samples. Microstructure seems to play an important role in the investigation under consideration.

Figure 8: extend the cation please, so that the meaning of each panel becomes clear. Also take a look at the axes titles which are very close to the panels in the area between them.

Altogether I see plenty of room for improvement of the paper. Especially when it comes to readability. Also keep in mind that the figures and their captions is what catches the reader’s attention. Finally please make clear the novelty of the study, it’s importance, applicability etc.

I recommend the publication after the manuscript is revised carefully.

Author Response

Dear reviewer and editor,

Thank you very much for your kind and responsible reviewing. The comments are of great importance to the improvement of our paper. After careful modifying and polishing, the revision has been finished. The response on the comments is uploaded in an attachment.

Best regards,

Pu Wang on behalf of all authors

Reviewer 2 Report

The work reported by Z. Zhu et al., “Study of the soft magnetic properties of FeSiAl magnetic powder cores by compounding with different content of epoxy resin”, is exciting and needs to be published. However, I am concerned about the reported work that needs to be revised before its publication.

1.      Does the lattice parameter change from the raw material (Sample P0) upon increasing the EP content (Sample P1-P6)? If yes, Give details.

2.      The authors should describe the process for the measurement of the densities of samples.

3.      Upon increasing the EP content, Ms value decreases slightly. Is it beneficial for the process?

4.      Please include the density of samples in table 1 for comparison.

Author Response

(The authors gave the same response as above.)

Reviewer 3 Report

1-     There are a lot of grammar and punctuation errors and mistakes. I highly suggest to check the whole manuscript carefully. It is better a person, who has better English knowledge, check the file.

2-     Add information of EP provider.

3-     Add Scherrer equation result into XRD characterization.

4-     The authors characterized the magnetic properties by AGFM. Why the Ms amount of P5 is larger than of P6? Also, why by adding more amount of EP, did magnetic property improv?

5-     The references do not have a same format.

Author Response

(The authors gave the same response as above.)

Reviewer 4 Report

The introduction should be dedicated to present critical analysis of state-of-the-art related work to justify the objective of the study. The authors must discuss along with the novelty of the present approach over the earlier one in a more elaborate way, not only an enumeration of literature data.

Section 2. Characterization (2.3) is difficult to follow. Please reconsider it for each sample type.

Results and discussion. In such a vast field, the discussion presented is poor, in terms of discussing its results and comparing them with the bibliography. I suggest reviewing this part more carefully and discuss further.

Section 2.3 What about the diffraction file?

Figure 3. What is Cum% on y axis?

The Figures should be revised/ uniformized in terms of size, font, etc.

Conclusions. Conclusion part should be rewritten to show what is the significance of your work for the study and to go beyond the results sections for forming the conclusions. What about the measurement unit of permeability?

The language of the manuscript must be significantly improved so that it is easy to read. You need to correct the grammar. Please go through the entire manuscript and shorten and correct some sentences. Please apply.

 In conclusion, considering the systematization and integrity of the methodology, as well as the writing quality of the manuscript, this paper cannot meet the requirements of publication in such a journal. Nevertheless, the efforts of performing all the experiments have been significant and I hope that in the near future all the issues will be solved.

Author Response

(The authors gave the same response as above.)

Round 2

Reviewer 2 Report

It has improved a lot, can be accepted 

Reviewer 3 Report

Dear Authors, 

Thank you for applying my comments to the file. 

Regards, 

Reviewer 4 Report

The manuscript can be published in the present form.